# Photosynthesis in Response to Different Salinities and Immersions of Two Native Rhizophoraceae Mangroves

**DOI:** 10.3390/cells11193054

**Published:** 2022-09-29

**Authors:** Chung-I Chen, Kuan-Hung Lin, Meng-Yuan Huang, Shau-Lian Wong, Tien-Szu Liao, Ming-Nan Chen, Jen-Hsien Weng, Mei-Li Hsueh, Yu-Hsiang Lai, Ching-Wen Wang

**Affiliations:** 1Department of Forestry, National Pingtung University of Science and Technology, Pingtung City 91201, Taiwan; 2Department of Horticulture and Biotechnology, Chinese Culture University, Taipei 11114, Taiwan; 3Department of Life Sciences, National Chung-Hsing University, Taichung 40227, Taiwan; 4Endemic Species Research Institute, Nantou 552005, Taiwan; 5Department of Forestry, National Chung Hsing University, Taichung 40227, Taiwan; 6Department of Agriculture, Taoyuan City Government, Taoyuan City 330206, Taiwan

**Keywords:** chlorophyll fluorescence, energy quenching, immersion, mangroves, photosynthesis, stomatal conductance

## Abstract

Mangrove ecosystems are vulnerable to rising sea levels as the plants are exposed to high salinity and tidal submergence. The ways in which these plants respond to varying salinities, immersion depths, and levels of light irradiation are poorly studied. To understand photosynthesis in response to salinity and submergence in mangroves acclimated to different tidal elevations, two-year-old seedlings of two native mangrove species, *Kandelia obovata* and *Rhizophora stylosa*, were treated at different salinity concentrations (0, 10, and 30 part per thousand, ppt) with and without immersion conditions under fifteen photosynthetic photon flux densities (PPFD μmol photon·m^−2^·s^−1^). The photosynthetic capacity and the chlorophyll fluorescence (ChlF) parameters of both species were measured. We found that under different PPFDs, electron transport rate (ETR) induction was much faster than photosynthetic rate (Pn) induction, and Pn was restricted by stomatal conductance (Gs). The Pn of the immersed *K. obovata* plants increased, indicating that this species is immersed-tolerant, whereas the Pn level of the *R. stylosa* plants is salt-tolerant with no immersion. All of the plants treated with 30 ppt salinity exhibited lower Pn but higher non-photochemical quenching (NPQ) and heat quenching (D) values, followed by increases in the excess energy and photoprotective effects. Since NPQ or D can be easily measured in the field, these values provide a useful ecological monitoring index that may provide a reference for mangrove restoration, habitat creation, and ecological monitoring.

## 1. Introduction

The family Rhizophoraceae includes the mangrove genera *Avicennia*, *Bruguiera*, *Ceriops*, *Kandelia*, *Lumnitzera*, and *Rhizophora* [1], and a diverse group of 81 taxa of mangroves belongs to 17 families and 30 genera in saline and tidal wetlands along tropical and subtropical coastlines [2]. The growth of mangroves is limited by temperature and exhibits a distinct geographical distribution, primarily from latitude 30° N to 30° S [3]. Generally, since mangroves are located in the intertidal zone, the environment is characterized by high evaporation [4] and high salinity variability [5]. Sabine et al. [6] revealed that mangroves have an optimal salinity range of 8–18 parts per thousand (ppt), but Krauss et al. [7] indicated an optimal seawater concentration ranged from 1.75 to 26.25 ppt, and the optimal salinity varies with species. Moreover, micro-tidal wetlands also show strong seasonal soil salinity variations (0–60 ppt) that may increase in amplitude based on climate prediction models, affecting the morphology and physiology of mangrove seedlings, thereby influencing the growth and species composition of mangrove swamps [4]. Previous studies have shown that salinities of 0–10 ppt are close to optimal for the growth of most mangroves [7,8]. Additionally, previous studies reported that the salt-tolerant mechanism related to salt stress might be induced by antioxidants, nitric oxide, HO1, and/or proline accumulation to minimize the effect of salinity on many plant species [9,10,11,12,13].

Among the four species of mangroves in Taiwan, *Kandelia obovata* Sheue Liu is adapted to areas with lower intertidal elevation and occupies the foreshore and offshore parts of littoral zones, growing at the middle zone of the mid-tidal line [14]. Thus, this species tolerates submergence [15]. Conversely, *Rhizophora stylosa* Griffith resides backshore to foreshore areas with relatively small tidal ranges and higher intertidal elevation [16]. Compared to *R. stylosa*, *K. obovata* seedlings at low elevations between the tides tend to increase their stem height and leaf numbers to capture light energy under the stimulation of immersion [17]. The same trend was also observed in the development of aerenchyma [18]. However, the changes in photosynthesis in response to salinity and immersion at various light irradiations of both *R. stylosa* and *K. obovata* acclimated to tidal elevation remain largely unknown. Eco-physiological responses to excess sunlight vary between different species and may significantly affect the survival rate and spatial distribution of the mangroves [19]. The saturation point of photosynthesis in mangroves is much lower than the luminosity that naturally occurs in its habitat, indicating that mangrove leaves often receive a large amount of excess light energy and may be prone to photo-inhibition [20,21]. In the harsh tropical intertidal environment of mangroves, the photosynthesis of mangroves can reach light saturation under the incident photon flux density of 40% sunlight (800–1000 μmol photon·m^−2^·s^−1^) or lower [22,23,24,25]. The difference in the photosynthesis capacity of mangroves is closely related to salinity. When mangrove ecosystems are exposed to high illumination and high salinity conditions, the environmental stresses may decrease stomatal conductance, resulting in reduced CO_2_ fixation to generate excess energy. The light protection and dissipation mechanism and ability of mangroves may affect plant performance and species distribution [5]. The high-illumination environment can also be a stress factor affecting halophytes. Therefore, photoprotection is a necessary feature for mangrove species to adapt to the intertidal environment [26]. In addition, as salinity increases, both net photosynthetic rate and stomatal conductance to water vapor decrease simultaneously [27]. The initial stage of photosynthetic induction resulted from photosynthesis, which had not completely started yet, leading to more light energy for photo-inhibition [28].

Threats to mangrove habitats include both the duration of future sea-level rises, subsequent high tide immersion [29], and changes in salinity [30]. It is necessary to determine the combined effects of immersion and increased salinity on mangrove seedlings for the most effective mangrove restoration [31]. Li et al. [32] reported that both Pn and gs of *A. marina*, a waterlog-tolerant mangrove, decreased after 8 h of inundation but increased after 4 h of inundation compared with the control (0 h of inundation). In addition, Wang et al. [33] indicated that *K. obovata* is more tolerant to waterlogging, and *R. stylosa* is more salt-tolerant. Evaluations of the photosynthetic responses of mangroves to salinity and immersion should help us to understand how the plants cope with fluctuations in light irradiance in the tropics and subtropics and to better predict how they will respond to anthropogenic climate change. We hypothesized that both salinity and immersion levels influence the photosynthetic characteristics, including the light energy absorption, utilization, dissipation, and photo-inhibition of *R. stylosa* and *K. obovata* mangroves in response to various light irradiances. Therefore, we designed three salinity and immersion experiments for *K. obovata* and *R. stylosa* seedlings. We aimed to elucidate the photosynthetic performances of both species treated with salinity and immersion under various light irradiations. The obtained data could explain differences in the distribution of mangrove species in the microenvironment and provide environmental conditions for mangrove restoration in the feature.

## 2. Materials and Methods

### 2.1. Plant Materials, Cultivation, and Salt Treatments

Both *K. obovata* and *R. stylosa* seedlings in the Tainan Shuangchun Estuary (TSE), Taiwan (23°17′38.8″ N, 120°06′42.3″ E) were collected, followed by planting in the nurseries at the Endemic Species Research Institute, Nantou, Taiwan (23°49′43.0″ N, 120°48′04.7″ E). This collection location represents the mangrove species’ natural range of planting those native mangrove species. These propagules are representative of the germplasms that would become established at the species’ Southern range limit. Similar size propagules without fungal infection and insect damage were cultivated using the clean sand collected from the intertidal zone habitat and placed in round Wagner pots (20 cm in diameter and 20 cm in height) for 2 years to simulate the natural growth environment. Those seedlings were grown at a greenhouse with an air temperature of 25–28 °C, a relative humidity of 50–70%, and a 12-h/12-h light/dark cycle with a photosynthetic photon flux density (PPFD) of 800–1500 μmol m^−2^·s^−1^. During the cultivation period, 15 ppt saline water was used, and 10 g of slow action fertilizer (total nitrogen 14%, water-soluble phosphoric acid 14%, potassium oxide 14%, and boron 0.05%) was applied every 3 months to ensure seedling growth and root development. Those seedlings with a height of 50–60 cm and a ground diameter of 3–4 cm were then selected for salinity and immersion treatments. The concentrations of the salinity treatments were 0 (fresh tap water, control), 10, and 30 ppt, with or without immersion treatment. The salinity treatments lasted for 6 months. In the immersion treatment, a plastic bucket with a diameter of 56 cm and a height of 75 cm (immersion height) is used for the whole plant to be submerged. The immersion treatments lasted for 7 days and 4 h daily. The salinity levels were obtained by mixing salt (ocean sea salt, Tainan City, Taiwan) with fresh tap water to attain the target salinity levels in the designated treatment buckets. The salinity concentrations were e-monitored weekly using a salinity sensor (PEN-SW, ATAGO, Tokyo, Japan), and new distilled water containing the target salinity was added as needed. The salinity concentrations were never more than 1–2 ppt higher or lower than target levels. A completely randomized design with three salinity treatments and two immersion treatments was used, and five replicates per treatment. Each treatment consisted of five seedlings (*n* = 5) of each species.

### 2.2. Measurement of Photosynthetic Capacity and Chlorophyll Fluorescence (ChlF) Parameters with a Fixed Light Source

After the salinity and immersion treatments, the PPFD was adjusted to 0, 5, 10, 15, 25, 50, 75, 100, 200, 400, 800, 1200, 1500, 1800, and 2000 μmol photon·m^−2^·s^−1^ in the leaf chamber to understand the radiant energy of the tested plants under various illumination conditions in the flow state. The artificial light source was provided by LED lights. The seedlings were measured with a gas-exchange and fluorescence photosynthesis analyzer (GFS-3000FL, Walz, Effeltrich, Germany) from April to June 2019 in the early summer to avoid the influence of temperature fluctuations. After the immersion of the seedlings in the morning, fully expanded and healthy leaves from the second paired leaf pair were dark-adapted for 30 min by the use of leaf clips. Following this, the central region of the adaxial leaf surface was subjected to a saturating light pulse of 3500 μmol m^−2^·s^−1^ (690 nm) prior to being measured. The values of the Fo and Fm of the dark-adapted samples were determined, and the gas exchange and ChlF measurements were simultaneously measured at 10:00 a.m. daily under the stable environmental conditions of the leaf chamber. The environmental conditions during the experiment were set to a gas-flow rate at 750 μmol s^−1^, gas-mixer speed to level 7, assimilator temperature to 25 °C, and relative humidity to 75%. All of the tested leaf samples were dry during the measurement to avoid experimental errors in gas exchange and ChlF indices. The chlorophyll fluorescence parameters we used are as follows, [28,34,35,36,37]:Fv/Fm = (Fm − Fo)/Fm
Fv’/Fm’ = (Fm’ − Fo’)/Fm’
ΔF/Fm’ = (Fm’ − F)/Fm’
ΦPSII = (Fm’ − Fs)/Fm’
ETR = ΦPSII × 0.5 × 0.84 × PPFD
ETR/PG = ETR/(Pn + R)
NPQ = (Fm/Fm’) − 1
D = 1 − (Fv’/Fm’)

The values for the net photosynthetic rate (Pn, μmol CO_2_ m^−2^·s^−1^) and stomatal conductance to water vapor (Gs, mmol H_2_O m^−2^·s^−1^) were simultaneously calculated and recorded inside the chamber of the photosynthesis analyzer (GFS-3000FL, Walz, Effeltrich, Germany). The operation was automatic, and the data were stored in the computer within the console and analyzed. All of the measurements were performed on fifteen leaves from five replicates for each treatment of 380–400 ppm in the atmospheric environment at room temperature (25 °C) from mid-morning until mid-afternoon (10:00~12:00).

### 2.3. Statistical Analysis

Statistical analyses were performed using PASW Statistics 18 software (PASW 18, IBM, USA). The gas exchange and ChlF measurements were analyzed using single-factor analysis of variance (ANOVA) to check for significant differences between *K. obovata* and *R. stylosa*. All of the PPFDs were arranged in a completely randomized design, and all ChlF parameters and gas exchange were subjected to a single-factor analysis of variance (ANOVA) to determine whether a significant difference level of *p* ≤ 0.05 (using PASW Statistics 18 software (PASW 18, IBM, USA) existed between *K. obovata* and *R. stylosa*. Multiple comparisons were performed using the least-significant difference. Regression analyses were used to examine relationships among ETR, ΦPSII, and NPQ. Those model datasets were based on at least 75 leaves from each PPFD level, and ChlF parameters were calculated using ETR data from the model validation datasets. Several models were tested, including the linear regression models being selected for the interpretation of the relationship between ChlF parameters and PPFD. All of the models were evaluated for the goodness of fit by the graphical analysis of the residuals and by computing correlation coefficients at a significance level of *p* ≤ 0.05 between the gas-exchange and ChlF parameters. The linear regression model performance was most suitable.

## 3. Results

### 3.1. Photosynthesis Curves of CO_2_ Fixation and Stomatal Conductance

The variations in the net photosynthetic rate (Pn) and stomatal conductance to water vapor (Gs) of the *K. obovata* and *R. stylosa* plants cultivated under salinity treatments with and without immersion conditions in response to PPFD illuminations are shown in Figure 1 and Figure 2. Significantly higher Pn values of *K. obovata* were detected under the 10 and 30 ppt salinity treatments at PPFD > 50 μmol·m^−2^·s^−1^ than those of the control (Figure 1a,c), and higher Pn was recorded when *K. obovata* plants were immersed (Figure 1c) than in the absence of immersion (Figure 1a) at illumination PPFD > 50 μmol·m^−2^·s^−1^. However, Pn levels of *R. stylosa* plants without immersion (Figure 1b) were higher than levels observed in immersed plants (Figure 1d) at > 200 μmol·m^−2^·s^−1^, and those plants treated with 10 ppt salinity under no immersion condition at 200–1800 PPFD had higher Pn levels than plants receiving the 30 ppt treatment and the control (Figure 1b).

At all PPFDs, higher Gs were recorded in *K. obovata* plants under immersion (Figure 2c) than in non-immersed plants (Figure 2a). Nevertheless, significantly lower Gs levels were found when *R. stylosa* plants were treated with 30 ppt salinity compared to the 10 ppt salinity treatment and the control at PPFD > 50 μmol·m^−2^·s^−1^ (Figure 2b,d).

The relationships between the Pn and Gs of the *K. obovata* and *R. stylosa* plants in different salinity treatments under 1200–2000 µmol PPFD m^−2^·s^−1^ illuminations are shown in Figure 3. Positive and significant r^2^ values (ranging from 0.61 to 0.99) were shown between the Pn and Gs of *K. obovata* and *R. stylosa*, indicating that the photosynthetic performance is controlled by the stoma, where the slope of the linear equation is Pn/Gs and equivalent to water-use efficiency (WUE). Notably, the same slope (0.11) was found in both linear equations (Figure 3c) between Pn and Gs of *K. obovata* under immersion treatments, suggesting that immersion would increase WUE. In Figure 3b,d, 30 ppt salinity treatments in *R. stylosa* increased WUE, but immersion reduced the WUE of both control and 10 ppt salinity treatments (0.03) compared to both under no immersion conditions (0.08).

### 3.2. Chlorophyll Fluorescence Parameters

The variations in Fv/Fm (Fv’/Fm’), ETR, ETR/PG, NPQ, and D of the *K. obovata* and *R. stylosa* plants cultivated under salinity treatments with and without immersion conditions in response to PPFD illuminations are shown in Figure 4 and Figure 5. The Fv/Fm (Fv’/Fm’) levels of the *K. obovata* plants under all treatments were significantly higher than the controls (Figure 4a), and these values slowly declined from 0.8 at 0 PPFD to 0.38 at 1200 PPFD, and remained stable at 0.38 when the illumination was 1200–2000 PPFDs. In addition, the Fv/Fm (Fv’/Fm’) levels of the *R. stylosa* plants also slowly declined from 0 to 1200 PPFD, followed by maintaining at similar levels from 1200 to 2000 PPFDs (Figure 4b). The ETR levels of the *K. obovata* plants under all treatments were significantly higher than the controls (Figure 4c), and these values rapidly increased from 0 to 400 PPFD, peaked at 800 PPFD, and then dropped thereafter. However, the ETR levels of the *R. stylosa* plants treated with 10 ppt salinity under no immersion condition from 1200 to 2000 PPFD were significantly higher than controls and other treatments (Figure 4d), and these values rapidly increased from 0 to 400 PPFD at salinity treatments without immersion, and then declined thereafter. These results indicate that *K. obovata* is more suitable for immersion treatment, while *R. stylosa* is not resistant to immersion and more suitable for 10ppt salinity treatment.

The NPQ values of the *K. obovata* plants under immersion conditions were higher than those without immersion, and these values of the immersed plants increased as PPFD increased and reached the peaks at 1500 PPFD (Figure 5a). Moreover, the NPQ levels of the *R. stylosa* plants treated with 30 ppt salinity under immersion were significantly higher than the controls and the other treatments, and these values rapidly increased from 0 to 1200 PPFD (Figure 5b). All of the D values of the *K. obovata* plants under control without immersion conditions were significantly higher than the other treatments, and these values increased as the PPFD increased, reaching a peak at 1500 PPFD (Figure 5c). In addition, the D values in the *R. stylosa* plants under all of the treatments and controls increased as the PPFD increased up to reach the peak at 1200 PPFD (Figure 5d).

Table 1 shows that ETR reached a peak at 800 µmol PPFD m^−2^·s^−1^. To understand the energy flow under both low and high lights, the correlations between ETR and the non-photochemical parameters (NPQ) were analyzed. Under low illumination treatments (0–400 µmol PPFD m^−2^·s^−1^), positive and significant r^2^ values (0.89–0.99) were detected between the ETR and NPQ of all of the tested plants under the controls and all treatments. However, at high illumination irradiations (800–2000 µmol PPFD m^−2^·s^−1^), negative and significant r^2^ values (0.85–0.90) were shown between the ETR and NPQ of the *K. obovata* plants under the controls and all treatments, except for under the control without immersion condition. No correlations were detected between the ETR and NPQ of the *R. stylosa* plants under the controls and all treatments in high illumination conditions.

## 4. Discussion

The light-response curves reveal the photosynthetic properties of plants and can be used to assess the efficiency of light utilization and optimal light conditions for plants, comprising fundamental aspects of plant eco-physiology and productivity research [38,39]. Stomata control gaseous exchange between the leaf interior and the external atmosphere and improve WUE for saving water [40]. Mangroves grow in the intertidal zone, and the salty soil hinders water absorption. Generally, it is not easy to obtain water in a salinity environment, so mangroves have evolved a series of adaptations to promote the water use efficiency of daytime photosynthesis to maintain the metabolic function in highly saline environments [41]. The stomatal opening speed for mangroves during photosynthetic induction is constrained, resulting in a conservative water-use strategy developed in response to the mangrove environment, which is similar to the high WUE characteristic of terrestrial xerophytes [42].

We attempted to examine the interactive effect of light intensities and salinities with immersions on the early development of two native mangrove species seedlings, and the results will provide environmental conditions for afforestation in *R. stylosa* and *K. obovata* plantations. The photosynthetic curves of *K. obovata* and *R. stylosa* began to enter the saturation stage as the illumination intensity approached 1200 PPFD, and positive and significant correlations were shown between Pn and Gs of the two mangroves under 1200–2000 PPFD (Figure 3). While ETR was saturated at 800 PPFD (Figure 4c,d), and NPQ at low light (0–800 PPFD) was in the rising state (Figure 4b and Figure 5a), indicating that the light energy absorbed by the *K. obovata* and *R. stylosa* seedlings at low light was higher compared to that required for carbon fixation, depending on the photoprotective mechanism to quench excess energy. These results demonstrate that changes in Pn were mainly affected by Gs during the photosynthesis saturation period, followed by maintaining high WUE. The *R. stylosa* plants treated with 30 ppt salinity had a higher WUE (slope = 0.13) than those treated with 20 ppt salinity and control (slope = 0.08 without immersion and 0.03 with immersion), suggesting that *R. stylosa* is a salt-tolerant species. In addition, higher Pn and Gs levels were detected in *K. obovata* plants under immersed conditions than in plants without immersion, indicating that *K. obovata* is an immersion-tolerant species. The Gs of *R. stylosa* in each treatment was lower than that of *K. obovata*, which might limit Pn, suggesting that Gs is the main limiting factor for mangrove photosynthesis. *R. stylosa* treated with 30 ppt salinity inhibited both Pn and Gs. In our study, under treatments of both 10 ppt and 30 ppt salinity, immersed *K. obovata* had significantly higher Pn than the control, whereas only the 30 ppt salinity treatments of non-immersed *R. stylosa* had a significant effect on Pn of *R. stylosa.* The Pn values of immersed *R. stylosa* were less than 5 μmol CO_2_·m^−2^·s^−1^, perhaps due to the closure of stomata. Therefore, when it is not immersed, the photosynthesis of *R. stylosa* is salt-tolerant, but the species is obviously not suitable for high-salinity immersed environments. This may be because the intertidal elevation of *R. stylosa* is higher than that of *K. obovata* [16]. The habitats of *R. stylosa* with higher intertidal elevations undergo greater salinity changes, and the salinity range of mangrove habitats on the Western coast of Taiwan is about 0.02 to 4.4% [43], and plants are less likely to be submerged [30]. Chen and Ye (2013) [44] found that when the seawater salinity was about 28‰, as the intertidal elevations decreased, Pn and Gs of *K. obovata* with salt intolerance decreased more significantly compared to the flood-resistant *A. marina*.

Photosynthetic performance is a good indicator of physiological stress levels in all plants, and the photosynthetic performance of mangroves reflects their adaptability to their habitats. The difference in the photosynthetic capacity of mangroves is related to salinity, and high-salinity stress affects the quenching mechanism and ability of photo-protection of plants and the distribution of the species [17]. Moreover, in mangrove leaves, ambient light is often above the light saturation point, so mangroves are often under conditions of excess light energy and may exhibit photo-inhibition [45]. Under high illumination, an excess of absorbed energy often leads to photo-inhibition and the reduced efficiency of photosynthesis. When plants encounter high light irradiances and the effect of carbon fixation is restricted, there are other electron receivers that replace CO_2_ to maintain the electron flow, and these photo-protective mechanisms prevent PSII from being damaged by light irradiances and counteract the harmful effects of excessive photon absorption [46]. NPQ is an important photo-protective mechanism of plants in response to the high light irradiance damage, which quenches excess energy and dissipates as heat to avoid the harmful effects of excessive photon absorption [47,48]. The increase in the xanthophyll cycle of NPQ is due to the activation of violaxanthin de epoxidase by the decrease in pH inside the thylakoids lumen; that is, the change in xanthophyll could change the PSII structure and organization toward a quenching state [49]. The increase in the xanthophyll cycle of NPQ could not be regenerated; electron transport became saturated and increased the possibility of photoinhibition. NPQ was then activated to avoid photoinhibition [50]. Our results can be applied to predict the changes in the physiology, population, and distribution of *R. stylosa* and *K. obovata.*

At the early stage of photosynthetic induction (<800 PPFD) in *K. obovata*, ETR increases more rapidly than photosynthesis, during which excess energy may be generated to form photo-inhibition [38]. Thus, the absorption of too many photons creates excess energy, which usually results in a reduction in the efficiency of the PSs [51]. When the illumination was at 1200–2000 PPFDs, Fv/Fm (Fv’/Fm’) and D of all tested plants were stably maintained at the low and high levels, respectively, with little or no change (Figure 4 and Figure 5), indicating that the photochemical ability had reached saturation. When *K. obovata* and *R. stylosa* were maintained at <800 PPFD, both Pn and NPQ increased simultaneously, suggesting that there is still an increase in excess light energy, potentially dissipated by D (Figure 1a,b and Figure 5a,b).

NPQ increased rapidly at 0–800 PPFD, implying that NPQ was mainly responsible for the dissipation of energy in the early stage of illumination for the *K. obovata* plants (Figure 5a). At 1500 PPFD, *K. obovata* under all treatments reached the saturation of NPQ, indicating that *K. obovata* is adapted to a high-illumination environment, and the changes of NPQ were closely related to Pn. However, at 1500–2000 PPFDs, NPQ levels were higher in immersed *K. obovata* than in non-immersed plants, indicating that the immersion had a great impact on *K. obovata*, and NPQ of this species was used to balance the excess irradiant energy associated with high illumination. At 1500–2000 PPFD, *K. obovata* still regulated excess energy with slightly increased levels of NPQ, but Pn began to decrease in high illumination (Figure 1a and Figure 5a). Under all treatments and illuminations, the ETR/PG ratio of *K. obovata* remained constant and close to 0, although ETR increased at 0–800 PPFD and decreased thereafter (Figure 4c,e), suggesting that electron acceptors other than CO_2_ were functioning, and NPQ was used as the photo-protective mechanism. The D values of *K. obovata* under the control condition, without immersion at 1200–2000 PPFD, were significantly higher than those under other treatments (Figure 5d), indicating that the heat quenching of *K. obovata* was associated with photo-protection and negative impacts on *K. obovata* were attributed to non-saline treatments. Thus, the elevated D level of *K. obovata* may help plants avoid high-illumination damage from excess energy.

NPQ = (Fm/Fm’ − 1) stands for non-photochemical quenching, which is divided into photoprotection and photoinhibition [33,34,35]. When the plant is below the saturation luminosity, Pn will raise and maintain the level. At this time, the NPQ rises to the effect of photoprotection. At high luminosity, Pn will decrease, and the NPQ rises to the effect of photoinhibition. In addition, when the total energy is treated as 1, the energy quenching can be divided into P, D, and E, where P = (Fm’ − Fs)/Fm’ is the ratio of photochemical quenching to absorbed energy; D = 1 − Fv’/Fm’ = (Fm’ − Fo’)/Fm, which is the ratio of thermal quenching to absorbed energy; E = 1 − P − D is the ratio of excess energy to the absorbed energy. The seedlings of *R. stylosa* had excess energy in low light because the rate of increase in Pn was slower than that of ETR (Figure 1b and Figure 4d), and at that time, both the ETR and D values of the *R. stylosa* plants reached saturation (Figure 4d and Figure 5d). Furthermore, *R. stylosa* seedlings increased heat quenching to cope with periods when Pn was low in low light, which correspond to the significant and positive correlations between ETR and NPQ of *R. stylosa* under all treatments at low illuminations (Table 1). These results indicate that the path of energy flow to NPQ was used mainly for photoprotection at this stage. When PPFD > 1500 µmol m^−2^·s^−1^, both Pn and ETR of *R. stylosa* decreased simultaneously (Figure 1b,d and Figure 4d), resulting in photo-inhibition and excess energy continued to rise, while D was still maintained in the maximum range of 0.65–0.75 (Figure 5d). Thus, *R. stylosa* seedlings in ETR reduced and maintained the heat quenching level to deal with the highlights. Both the NPQ and D values of *R. stylosa* treated with 30 ppt salinity were higher than those of other treatments and controls (Figure 5b,d), indicating that high salinity has an adverse impact on *R. stylosa*, and NPQ or D may be further elevated to avoid the damage of excess light energy [38]. The excess light energy generated would accordingly maintain D and NPQ (Figure 5d) due to the photo-protective mechanism continuously maintained by the presence of a high proportion of NPQ as the illumination increased. These results can be applied to predict photosynthetic and respiratory responses to light irradiances in young mangroves.

## 5. Conclusions

Both species showed different capacities for acclimating to light irradiations, exhibiting protective mechanisms to avoid damage to the photosynthetic apparatus. The Pn level of *K. obovata* was not elevated when the plants were immersed, whereas the Pn of the non-immersed *R. stylosa* plants was more salt-tolerant than *K. obovata* without immersion. These photosynthetic performances of both species reflected their adaptability to their environments. Generally, the photosynthetic activity of *K. obovata* was suitable in both the 10 and 30 ppt salinity treatments, whereas *R. stylosa* performed well at only 10 ppt salinity. Both NPQ and Pn increased as PPFD increased, but Pn decreased when PPFD was >1500 μmol photon m^−2^·s^−1^, indicating the adjustment of both species to acclimate to dynamic changes in light conditions. Under different luminosity gradients, ETR induction was much faster than Pn induction, resulting in excess energy production and the use of NPQ for the dissipation of excessive energy. In all plants treated with 30 ppt salinity at >1500 PPFD, Pn levels were reduced while NPQ and D values were elevated, leading to enhanced photoprotection. The NPQ and D values are easy to measure in the field and may possibly be used as ecological monitoring indicators to provide a reference for mangrove restoration and habitat construction.

## Figures and Tables

**Figure 1 cells-11-03054-f001:**
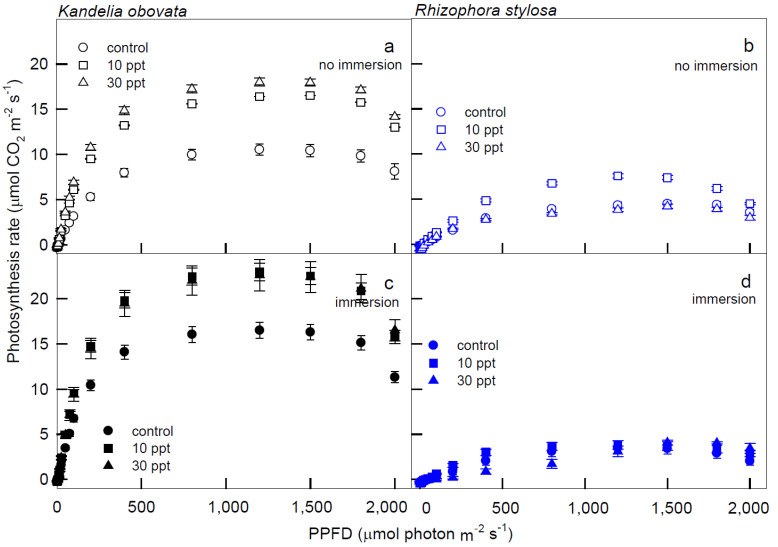
The light−response curves of net photosynthetic rate (Pn) in *K. obovata* (**a**,**c**) and *R. stylosa* (**b**,**d**) seedlings cultivated under 0 ppt (fresh tap water, control), 10 ppt, and 30 ppt of salinity without immersion (**a**,**b**) and with immersion (**c**,**d**) conditions. Measurements were recorded at 25 °C with different photosynthetic photon flux density (PPFD) at 0, 5, 10, 15, 25, 50, 75, 100, 200, 400, 800, 1200, 1500, 1800, and 2000 μmol photon·m^−2^·s^−1^. Each data point represents as mean ± standard error (SE) of 3 leaves from five replicates (*n* = 15) for each salinity treatment.

**Figure 2 cells-11-03054-f002:**
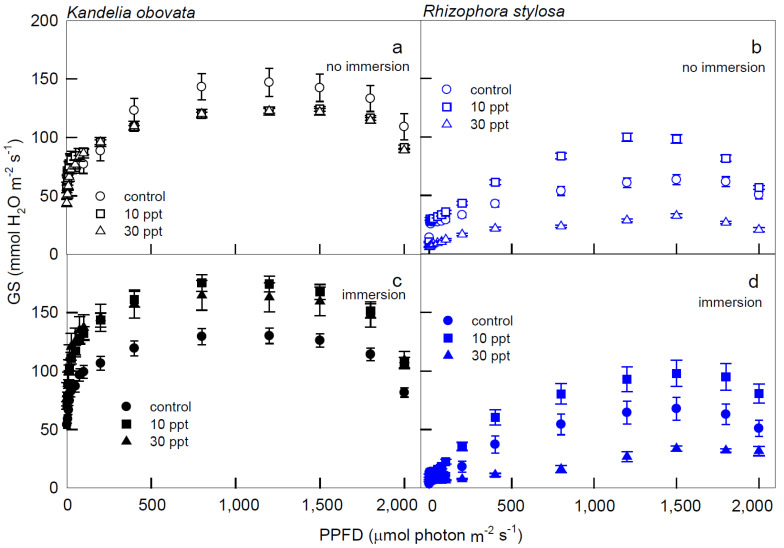
The light−response curves of stomatal conductance (Gs) in *K. obovata* (**a**,**c**) and *R. stylosa* (**b**,**d**) seedlings cultivated under 0, 10, and 30 ppt of salinity without immersion (**a**,**b**) and with immersion (**c**,**d**) conditions. Measurements were recorded at 25 °C with different PPFD at 0, 5, 10, 15, 25, 50, 75, 100, 200, 400, 800, 1200, 1500, 1800, and 2000 μmol photon·m^−2^·s^−1^. Each data point represents as mean ± SE of 3 leaves from five replicates (*n* = 15) for each salinity treatment.

**Figure 3 cells-11-03054-f003:**
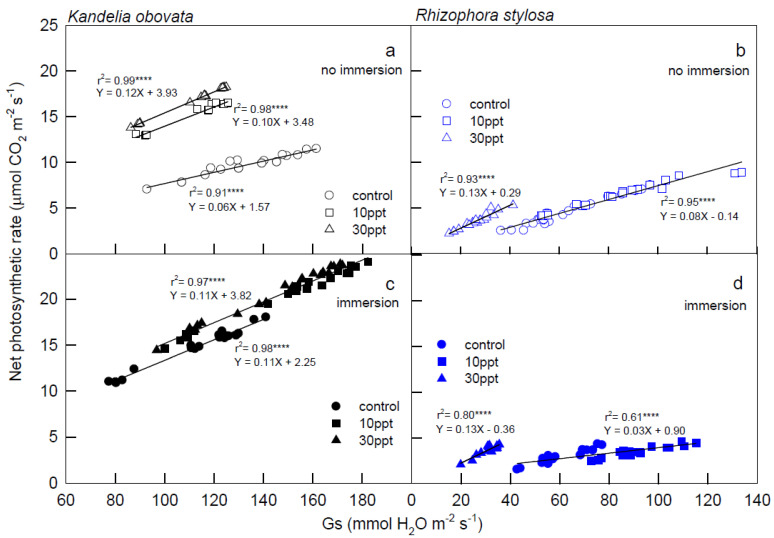
The relationship between stomatal conductance (Gs) and net photosynthetic rate (Pn) of *K. obovata* (**a**,**c**) and *R. stylosa* (**b**,**d**) cultivated under 0, 10, and 30 ppt salinity treatments with (**c**,**d**) and without (**a**,**b**) immersion conditions. Measurements were recorded at 25 °C and 1200−2000 μmol·m^−2^·s^−1^ PPFD. Each symbol represents the average of one leaf on one seedling, and five seedlings were randomly selected from each treatment under 1200, 1500, 1800, 2000 PPFD luminosity changes, and a total of 20 values were used for each linear regression analysis. Determination coefficient (r^2^) provides a measure of regression model fit, and the model *p*-values indicate model significance at *p*
*<* 0.0001, denoted with ****.

**Figure 4 cells-11-03054-f004:**
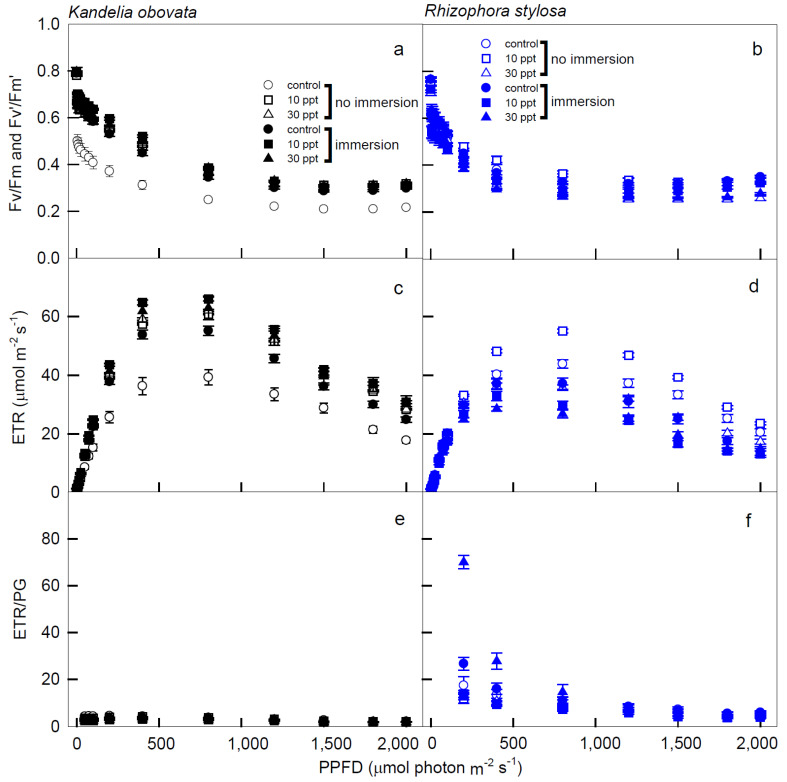
The light−response curves of Fv/Fm (Fv’/Fm’), electron transport rate (ETR), and electron transport rate/gross photosynthesis (ETR/PG) in *K. obovata* (**a**,**c**,**e**) and *R. stylosa* (**b**,**d**,**f**) seedlings cultivated under 0, 10, and 30 ppt of salinity with and without immersion conditions. Measurements were recorded at 25 °C with different PPFD at 0, 5, 10, 15, 25, 50, 75, 100, 200, 400, 800, 1200, 1500, 1800, and 2000 μmol photon·m^−2^·s^−1^. Each data point represents the mean ± SE of 3 leaves from five replicates (*n* = 15) for each salinity treatment.

**Figure 5 cells-11-03054-f005:**
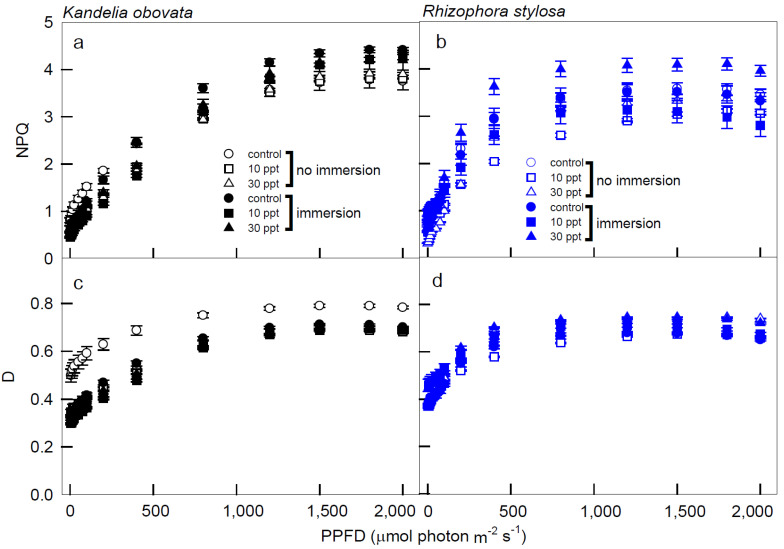
The light−response curves of non-photochemical quenching (NPQ) and heat quenching (D) in *K. obovata* (**a**,**c**) and *R. stylosa* (**b**,**d**) seedlings cultivated under 0, 10, and 30 ppt of salinity with and without immersion conditions. Measurements were recorded at 25 °C with different PPFD at 0, 5, 10, 15, 25, 50, 75, 100, 200, 400, 800, 1200, 1500, 1800, and 2000 μmol photon·m^−2^·s^−1^. Each data point represents a mean ± SE of three leaves from five replicates (*n* = 15) for each salinity treatment.

**Table 1 cells-11-03054-t001:** **The relationship****between ETR and NPQ** of *K. obovata* and *R. stylosa* cultivated under 0, 10, and 30 ppt salinity treatments with and without immersion conditions. Measurements were recorded at 25 °C with PPFD at 0–400 (low) and 800–2000 (high) μmol photon·m^−2^·s^−1^. Each symbol represents the average of 5 seedlings randomly selected from each treatment under 0, 5, 10, 15, 25, 50, 75, 100, 200, 400, 800, 1200, 1500, 1800, and 2000 PPFD luminosity changes, and a total of 5–10 values were used for each linear regression analysis.

			*K. obovata*	*R. stylosa*
			ETR × NPQ	ETR × NPQ
PPFD	Immersion	Salinity	R^2^ with Positive (+) or Negative (-) Correlation	
Low (0–400)	no immersion	control	0.99 **** (+)	0.98 **** (+)
		10 ppt	0.99 **** (+)	0.99 **** (+)
		30 ppt	0.98 **** (+)	0.90 **** (+)
	immersion	control	0.99 **** (+)	0.92 **** (+)
		10 ppt	0.98 **** (+)	0.91 **** (+)
		30 ppt	0.98 **** (+)	0.89 **** (+)
High (800–2000)	no immersion	control	0.75 (-)	0.14 (-)
		10 ppt	0.85 * (-)	0.76 (-)
		30 ppt	0.87 * (-)	0.66 (-)
	immersion	control	0.85 * (-)	0.13 (+)
		10 ppt	0.90 * (-)	0.41 (+)
		30 ppt	0.85 * (-)	0.003 (-)

Determination coefficient (R^2^) provides a measure of regression model fit, and the model *p*-values indicate model significance at *p* < 0.0001, and < 0.05 denoted with **** and *.

## Data Availability

The data and materials are available upon reasonable request from the corresponding author.

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
