# Peer review of "Photosynthesis in Response to Different Salinities and Immersions of Two Native Rhizophoraceae Mangroves"

_cells, 2022, doi:10.3390/cells11193054_

Round 1

Reviewer 1 Report

Comments to the Author

The manuscript entitled “Photosynthesis in response to different salinities and immersions at light irradiance gradients of two native Rhi-zophoraceae mangroves”.

The study is scientifically attractive and interesting manuscript on important species of mangroves which will draw attention of wide audience and researchers related working

The overall quality of the presented manuscript is adequate; however some minor corrections and revisions required.

Following are some specific suggestions which might help authors to improve the manuscript.

Introduction

Introduction section can be elaborated with inclusion of recent reference of salt stress and its tolerance mechanism and important role of some antioxidant enzyme and some recently identified enzyme HO-1 and Nitric oxide, following are some suggestion worth to include on salt induced stress and its tolerance mechanism.  

·         Khator et al (2020). NaCl induced oxidative stress in legume crops of Indian Thar Desert: an insight in the cytoprotective role of HO 1, NO and antioxidants. Physiology and Molecular Biology of Plants. 26: 51–62

·         Khushboo Khator and G. S. Shekhawat (2019). Nitric oxide improved salt stress tolerance by osmolyte accumulation and activation of antioxidant defense system in seedling of B. juncea (L.) Czern. Vegetos. 32: 583-592 doi.10.1007/s42535-019- 00071-

·         Lovely Mahawar and G. S. Shekhawat (2019). EsHO 1 mediated mitigation of NaCl induced oxidative stress and correlation between ROS, antioxidants and HO 1 in seedlings of Eruca sativa: Underutilized oil yielding crop of arid region. Physiology and Molecular Biology of Plants. 25 (4):895–904

·         Mahawar, L, G. S. Shekhawat (2018). Haem oxygenase: a functionally diverse enzyme of photosynthetic organisms and its role in Phytochrome chromophore biosynthesis, cellular signalling and defence mechanisms. Plant cell and Environment 41:483–500 DOI: 10.1111/pce.13116

·         Mahawar et al. (2018). Role of Proline in mitigating NaCl induced oxidative stress in Eruca sativa Miller: an important oil yielding crop of Indian Thar Desert. Vegetos 31: 1-6

Use either abbreviations or full scientific names of plants throughout the manuscript.

Author Response

The manuscript entitled “Photosynthesis in response to different salinities and immersions at light irradiance gradients of two native Rhi-zophoraceae mangroves”. The study is scientifically attractive and interesting manuscript on important species of mangroves which will draw attention of wide audience and researchers related working.

The overall quality of the presented manuscript is adequate; however some minor corrections and revisions required.Following are some specific suggestions which might help authors to improve the manuscript.

Introduction

Introduction section can be elaborated with inclusion of recent reference of salt stress and its tolerance mechanism and important role of some antioxidant enzyme and some recently identified enzyme HO-1 and Nitric oxide, following are some suggestion worth to include on salt induced stress and its tolerance mechanism.

*Response: *Thank you very much for your positive comments. As suggested, all mentioned recent references of salt stress and its tolerance mechanism have been added to the text in Line 49-52. __

“Additionally, previous studies reported that the salt tolerant mechanism related to salt stress may be induced byantioxidants, nitric oxide, HO1 and/or proline accumulation to minimize the effect of salinity on many plant species [9, 10, 11, 12, 13].”

[9] Khator et al (2020). NaCl induced oxidative stress in legume crops of Indian Thar Desert: an insight in the cytoprotective role of HO 1, NO and antioxidants. Physiology and Molecular Biology of Plants. 26: 51–62.·

[10] Khushboo Khator and G. S. Shekhawat (2019). Nitric oxide improved salt stress tolerance by osmolyte accumulation and activation of antioxidant defense system in seedling of B. juncea (L.) Czern. Vegetos. 32: 583-592. doi.10.1007/s42535-019- 00071-

[11] Lovely Mahawar and G. S. Shekhawat (2019). EsHO 1 mediated mitigation of NaCl induced oxidative stress and correlation between ROS, antioxidants and HO 1 in seedlings of Eruca sativa: Underutilized oil yielding crop of arid region. Physiology and Molecular Biology of Plants. 25 (4):895–904.

[12] Mahawar, L, G. S. Shekhawat (2018). Haem oxygenase: a functionally diverse enzyme of photosynthetic organisms and its role in Phytochrome chromophore biosynthesis, cellular signalling and defence mechanisms. Plant cell and Environment 41:483–500. DOI: 10.1111/pce.13116.

[13] Mahawar et al. (2018). Role of Proline in mitigating NaCl induced oxidative stress in Eruca sativa Miller: an important oil yielding crop of Indian Thar Desert. Vegetos 31: 1-6.

Use either abbreviations or full scientific names of plants throughout the manuscript.

*Response:*Abbreviations of */K/**/./**/obovata/*and */R/**/./**/stylosa/*for/Kandelia obovata/ and /Rhizophora////stylosa/have been used throughout the revised manuscript.

Reviewer 2 Report

The authors compare the photosynthetic performance of two mangrove species as a function of salinity and light intensity. The outcomes of the study were as expected high light induces stress and very low salinity impairs photosynthetic performance. This paper provides no new novel insights but is an accounting of performance characteristics and as such is not appropriate for this journal,. The subject has a better fit for a journal focusing on physiological ecology. There were also a number of statements that were biologically incorrect including statements made on lines 70 and 77 regarding light absorption and photosynthesis not starting. Finally, the English in the manuscript requires substantial revision. The poor English substantially reduces the quality of the research.

Author Response

The authors compare the photosynthetic performance of two mangrove species as a function of salinity and light intensity. The outcomes of the study were as expected high light induces stress and very low salinity impairs photosynthetic performance. This paper provides no new novel insights but is an accounting of performance characteristics and as such is not appropriate for this journal,. The subject has a better fit for a journal focusing on physiological ecology. There were also a number of statements that were biologically incorrect including statements made on lines 70 and 77 regarding light absorption and photosynthesis not starting. Finally, the English in the manuscript requires substantial revision. The poor English substantially reduces the quality of the research.

Response: Mangroves ecosystems are vulnerable to rising sea levels as the plants are exposed to high salinity and tidal submergence. The ways in which these plants respond to varying salinities, immersion depths, and levels of light irradiation is poorly studied. To understand the photosynthesis in response to salinity and submergence in mangroves acclimated to different tidal elevations, two-year-old seedlings of two native mangrove species, Kandelia obovata and Rhizophora stylosa, were treated at different salinity concentrations (0, 10, 30 ppt) with and without immersion conditions under fifteen light irradiant gradients of μmol photon·m-2·s-1. Photosynthetic capacity and chlorophyll fluorescence parameters of both species were measured. The obtained data could explain differences in the distribution of mangrove species in the microenvironment and be used as ecological monitoring indicators to provide a reference for mangrove restoration and habitat construction.

Line 70 (old): The statement “When mangrove ecosystems are exposed to high illumination and high salinity conditions, the plants increase solar energy absorption and decrease stomatal conductance, resulting in reduced CO2 fixation to generate excess energy.” has been changed. The revised one now reads as follow.

“When mangrove ecosystems are exposed to high illumination and high salinity conditions, the environmental stresses may decrease stomatal conductance, resulting in reduced CO2 fixation to generate excess energy.” (Line 71-73)

Line 77 (old): The statement “The initial stage of photosynthetic induction was used because photosynthesis had not completely started, and a higher photosynthetic rate consumed more light energy and reduced photo-inhibition [23].” has been changed. The revised one now reads as follow.

“The initial stage of photosynthetic induction resulted from photosynthesis that had not completely started yet, leading to more light energy for the photo-inhibition [28].”  (Line 79-81)

Furthermore, the manuscript have been edited and reviewed by a native English-speaking colleague, Mr. Buford Pruitt ([email protected]), whose major is Biology.

Reviewer 3 Report

The paper by Chen and co-workers want to analyze how the increase in salinity and the plant immersion affect the photosynthetic performance of two different mangrove genera. The work is interesting and provide some useful insight on how these plants react to stress but, in my opinion, the work needs to be improved to be accepted.

In title and abstract, authors specify that plants were analyzed under fifteen light irradiant gradients of photosynthetic photon flux density. But the different light intensities are a tool to obtain a light curve and estimate plant photosynthetic response. This paper is about response to different salinity and immersion conditions and not a study of plants adapted at different light intensities. I suggest eliminating “at light irradiance gradients” from the title and from the description in the abstract.

In this work, some of the plants are immersed and some not. Is it the light that reach the immersed samplesw identical to the one for the control? Is it the spectra for the different samples identical since red light have a lower penetration in water? If the plants are adapted two different light intensities with different spectra this will clearly influence the photosynthetic response at the different light intensities.

Photoinhibition is discussed and it is always related to NPQ but it is never measured in this work. While NPQ is correctly measured there are no quantification of photoinhibiton for the different light intensities.  Simplest way to do this is to monitor Fv/Fm recover in the dark after illumination at the different lights intensities. The kinetic of recovery will also allow to distinguish between regulated (PSII supercomplex disassemble) or unregulated (photodamage and chlorophyll bleaching) PSII inhibition since the two phenomena have different time of recovery.

In figure 4 K. obovate showed an Fv/Fm in the control lower than in all the other treatment. How do the authors explain this? Why is the PSII yield lower in control condition than increasing salt concentration or after immersion? Is it again a problem of the light that reach the sample?

Line 352: this sentence is wrong. “The increase in the xanthophyll cycle of NPQ is caused by the change in the structure of the PSII antenna system; that is, the change in the rate of heat dissipation of excess light energy” The change in xanthophyll composition is due to activation of violaxanthin de epoxidase by the decrease of pH inside thylakoids lumen. Lumen acidification is due the excess light energy absorption and saturate photochemistry.  It‘s the change in xanthophyll that then could change the PSII structure and organization toward a dissipative state (NPQ). 

Line 309 “The photosynthetic induction of K. obovata took more than 1 h to achieve a high Pn level at 800 PPFD, and the rise of PSII efficiency was faster than that of CO2 fixation efficiency during photosynthetic induction.” Leaves are illuminated for 15 minutes at the different light intensity. Where the data for one hour at 800 PPFD come from? Do the authors mean the cumulative time of illumination at the different light intensities? In this case, they could not say that they need more than an hour at 800 to achieve high Pn. Besideds, during illumination PSII efficiency decrease and do not rise.

Line 354: “When the dissipation capacity is close to saturation, the excess electrons gradually begins to carry out the Calvin-Benson cycle of carbon fixation to prevent PSII from being damaged by light” this is wrong too. Calvin-Benson cycle is always active not only when dissipation mechanism are saturated. It’s just the opposite. When rubisco activity became limiting and NADPH could not be regenerated, electron transport became saturated and this increase the possibility of photoinhibiton. NPQ is then activate (when photochemistry became saturated) to avoid photoinhibition.

Table 1. the correlation between ETR and ΦPSII is strange. ETR is calculated as ETR = ΔF/Fm’ x PPFD x 0.5 x 0.84. ΔF/Fm’ = Fm’- f0’/Fm’ = photosystem II efficiency (ΦPSII). So ETR = ΦPSII x PPFD x 0.5 x 0.84. Basically, to obtain ETR the software multiplies ΦPSII for values that are fixed for all the leaves.   I do not think that correlation between these two parameters is useful since ETR is already by definition correlated to PSII efficiency.

Parameter D (defined as heat quenching) is never well explained.  In this work Parameter D is reported but it is never discussed and distinguished from NPQ. In material and methods it is stated: “Nonphotochemical quenching and its components are represented in the following equations: non-photochemical quenching (NPQ) = Fm /Fm’ – 1 and D = 1- (Fv’/Fm’).  1- (Fv’/Fm’) is not a measure of an NPQ component. It indicates the fraction of light energy absorbed that is not used for photochemistry. It comprehends NPQ but also fluorescence emission and triplets quenching.  Authors should discuss in detail the difference between NPQ and D and what these differences represent

Line 69: “When mangrove ecosystems are exposed to high illumination and high salinity conditions, the plants increase solar energy absorption and decrease stomatal conductance, resulting in reduced CO2 fixation to generate excess energy.” This sentence is misleading; it’s not correct that in high light and reduced CO2 more excess energy is generated. In high light and reduced CO2, the electron transport became saturated and thus the light energy in excess could generate photooxidative stress.

Minor

Line 42-49: salinity is provided with three different units: percentage, millimole and part per thousand. Please use a common unit.

Line 67-68: “the photosynthesis of mangroves can reach light saturation under the incident photon flux density of 40% sunlight or lower”. Please provide an approximate value, of the light intensity that saturate photosynthesis in mangrove.

Author Response

The paper by Chen and co-workers want to analyze how the increase in salinity and the plant immersion affect the photosynthetic performance of two different mangrove genera. The work is interesting and provide some useful insight on how these plants react to stress but, in my opinion, the work needs to be improved to be accepted.

Response: Thank you very much for your positive comments.

In title and abstract, authors specify that plants were analyzed under fifteen light irradiant gradients of photosynthetic photon flux density. But the different light intensities are a tool to obtain a light curve and estimate plant photosynthetic response. This paper is about response to different salinity and immersion conditions and not a study of plants adapted at different light intensities. I suggest eliminating “at light irradiance gradients” from the title and from the description in the abstract.

Response: These words “at light irradiance gradients” have been eliminated from the title and from the description in the abstract as suggested.

In this work, some of the plants are immersed and some not. Is it the light that reach the immersed samples identical to the one for the control? Is it the spectra for the different samples identical since red light have a lower penetration in water? If the plants are adapted two different light intensities with different spectra this will clearly influence the photosynthetic response at the different light intensities.

Response: Wang et al. [33] indicated that K. obovata is more tolerant to waterloging and R. stylosa is more salt-tolerant. Evaluations of the photosynthetic responses of mangroves to salinity and immersion should help us to understand how the plants cope with fluctuations in light irradiance in the tropics and subtropics, and to better predict how they will respond to anthropogenic climate change. We hypothesized that both salinity and immersion levels influence the photosynthetic characteristics, including the light energy absorption, utilization, dissipation, and phot-inhibition, of R. stylosa and K. obovata mangroves in response to various light irradiances. (L88-95)

The following sentence has been added to Materials and methods to clarify the ’sample” issue.

“All tested leaf samples were dry during the measurement to avoid experimental errors in gas exchange and chlorophyll fluorescence (ChlF) indices. (Line 148-149).”

Photoinhibition is discussed and it is always related to NPQ but it is never measured in this work. While NPQ is correctly measured there are no quantification of photoinhibiton for the different light intensities.  Simplest way to do this is to monitor Fv/Fm recover in the dark after illumination at the different lights intensities. The kinetic of recovery will also allow to distinguish between regulated (PSII supercomplex disassemble) or unregulated (photodamage and chlorophyll bleaching) PSII inhibition since the two phenomena have different time of recovery.

Response: Thank you very much for your suggestions on the evaluation of composition of NPQ (qE, qZ + qT, qZ) using the different light intensities of light induction, which can clearly understand the relationship between photoprotection and photoinhibition as described by Wang et al. [33].

In the study, we tried to identify the differences between K. obovata and R. stylosa, so we aimed to illustrate the relationship between gas exchange and chlorophyll fluorescence in the light response curve of K. obovata and R. stylosa under the context of different photometric gradients using the reaction of Pn and NPQ of seedlings. When the plants were below the saturation luminosity, Pn values would rise and maintain the level, and NPQ also rose to the effect of photoprotection at this time as well. Nevertheless, at high luminosity, Pn values would decrease and NPQ rose to the effect of photoinhibition. These results demonstrate the diverse roles of NPQ.

In figure 4 K. obovate showed an Fv/Fm in the control lower than in all the other treatment. How do the authors explain this? Why is the PSII yield lower in control condition than increasing salt concentration or after immersion? Is it again a problem of the light that reach the sample?

Response: Mangroves are halophytes and grow in the area where seawater and freshwater meet, especially to the species of K. obovata that is adapted to immersion and salinity. Fv/Fm is the maximum potential of the un-illuminated photosynthetic system, whereas PSII is the efficiency of the photosynthetic system in the illumination. Decreased values of both Fv/Fm and PSII have been observed when K. obovata plants grow under unsuitable growth environments. Thus, we have added the following sentence to Materials and methods to clarify the ’sample” issue.

“All tested leaf samples were dry during the measurement to avoid experimental errors in gas exchange and chlorophyll fluorescence (ChlF) indices. (Line 148-149).”

Line 352: this sentence is wrong. “The increase in the xanthophyll cycle of NPQ is caused by the change in the structure of the PSII antenna system; that is, the change in the rate of heat dissipation of excess light energy” The change in xanthophyll composition is due to activation of violaxanthin de epoxidase by the decrease of pH inside thylakoids lumen. Lumen acidification is due the excess light energy absorption and saturate photochemistry.  It‘s the change in xanthophyll that then could change the PSII structure and organization toward a dissipative state (NPQ). 

Response: The statement “The increase in the xanthophyll cycle of NPQ is caused by the change in the structure of the PSII antenna system; that is, the change in the rate of heat dissipation of excess light energy [44].” has been modified. The revised one now reads as follow.

 “The increase in the xanthophyll cycle of NPQ is due to activation of violaxanthin de epoxidase by the decrease of pH inside thylakoids lumen; that is, the change in xanthophyll could change the PSII structure and organization toward a dissipation state [49].” (Line 392-396)

Line 309 “The photosynthetic induction of K. obovata took more than 1 h to achieve a high Pn level at 800 PPFD, and the rise of PSII efficiency was faster than that of CO2 fixation efficiency during photosynthetic induction.” Leaves are illuminated for 15 minutes at the different light intensity. Where the data for one hour at 800 PPFD come from? Do the authors mean the cumulative time of illumination at the different light intensities? In this case, they could not say that they need more than an hour at 800 to achieve high Pn. Besideds, during illumination PSII efficiency decrease and do not rise.

Response: The statement “The photosynthetic induction of K. obovata took more than 1 h to achieve a high Pn level at 800 PPFD, and the rise of PSII efficiency was faster than that of CO2 fixation efficiency during photosynthetic induction. After photosynthesis was induced by different levels of illumination, Pn of each species was closely related to Gs. Photosynthetic curves of K. obovata and R. stylosa began to enter the saturation stage as the illumination intensity approached 1,200 PPFD, and positive and significant correlations were shown between Pn and Gs of the two mangroves under 1,200 - 2,000 PPFD.” has been modified. The revised one now reads as follow.

“Photosynthetic curves of K. obovata and R. stylosa began to enter the saturation stage as the illumination intensity approached 1,200 PPFD, and positive and significant correlations were shown between Pn and Gs of the two mangroves under 1,200 - 2,000 PPFD (Fig. 3). While ETR was saturated at 800 PPFD (Figs. 4c, d), and NPQ at low light (0 -800 PPFD) was in the rising state (Fig. 5a, Fig. 4b), indicating that the light energy absorbed by K. obovata and R. stylosa seedlings at low light was higher compared to required for carbon fixation, depending on photoprotective mechanism to quench excess energy.” (L348-355)

Line 354: “When the dissipation capacity is close to saturation, the excess electrons gradually begins to carry out the Calvin-Benson cycle of carbon fixation to prevent PSII from being damaged by light” this is wrong too. Calvin-Benson cycle is always active not only when dissipation mechanism are saturated. It’s just the opposite. When rubisco activity became limiting and NADPH could not be regenerated, electron transport became saturated and this increase the possibility of photoinhibiton. NPQ is then activate (when photochemistry became saturated) to avoid photoinhibition.

Response: The statement “When the dissipation capacity is close to saturation, the excess electrons gradually begins to carry out the Calvin-Benson cycle of carbon fixation to prevent PSII from being damaged by light [45].” has been modified. The revised one now reads as follow.

“When rubisco activity became limiting and NADPH could not be regenerated, electron transport became saturated and increase of the possibility of photoinhibition. NPQ was then activated to avoid photoinhibition [50].”  (Line 394-397)

Table 1. the correlation between ETR and ΦPSII is strange. ETR is calculated as ETR = ΔF/Fm’ x PPFD x 0.5 x 0.84. ΔF/Fm’ = Fm’- f0’/Fm’ = photosystem II efficiency (ΦPSII). So ETR = ΦPSII x PPFD x 0.5 x 0.84. Basically, to obtain ETR the software multiplies ΦPSII for values that are fixed for all the leaves.   I do not think that correlation between these two parameters is useful since ETR is already by definition correlated to PSII efficiency.

Response: The statement related to Table 1 has been modified, and the revised one now reads as follow.

“Table 1 shows that ETR reached a peak at 800 µmol PPFD m-2 s-1. To understand the energy flow under both low and high lights, the correlations between ETR and non-photochemical parameters (NPQ) were analyzed. Under low illumination treatments (0 – 400 µmol PPFD m-2 s-1), positive and significant r2 values (0.89 – 0.99) were detected between ETR and NPQ of all tested plants under controls and all treatments. However, at high illumination irradiations (800 - 2,000 µmol PPFD m-2 s-1), negative and significant r2 values (0.85 – 0.90) were shown between ETR and NPQ of K. obovata plants under controls and all treatments, except for under control without immersion condition. No correlations were detected between ETR and NPQ of R. stylosa plants under controls and all treatments in high illumination conditions.”    (Line 288-297)

Table 1. The relationship between ETR and NPQ of K. obovata and R. stylosa  cultivated under 0, 10, and 30 ppt salinity treatments with and without immersion conditions. Measurements were recorded at 25 °C with PPFD at 0 - 400 (low) and 800 - 2,000 (high) μmol photon·m-2·s-1. Each symbol represents the average of 5 seedlings randomly selected from each treatment under 0, 5, 10, 15, 25, 50, 75, 100, 200, 400, 800, 1,200, 1,500, 1,800, and 2,000 PPFD luminosity changes, and a total of 5-10 values were used for each linear regression analysis.

K. obovata

R. stylosa

ETR × NPQ

 ETR × NPQ

PPFD

immersion

salinity

R2 with positive(+) or negative(-) correlation

Low

(0 - 400)

no immersion

control

0.99****(+)

0.98****(+)

10 ppt

0.99****(+)

0.99****(+)

30 ppt

0.98****(+)

0.90****(+)

immersion

control

0.99****(+)

0.92****(+)

10 ppt

0.98****(+)

0.91****(+)

30 ppt

0.98****(+)

0.89****(+)

High

(800 - 2000) 

no immersion

control

0.75 (-)

0.14 (-)

10 ppt

0.85* (-)

0.76 (-)

30 ppt

0.87* (-)

0.66 (-)

immersion

control

0.85* (-)

0.13 (+)

10 ppt

0.90* (-)

0.41 (+)

30 ppt

0.85* (-)

0.003 (-)

Determination coefficient (R2) provides a measure of regression model fit, and the model p-values indicate model significance at p < 0.0001, and < 0.05 denoted with **** and *.

Parameter D (defined as heat quenching) is never well explained.  In this work Parameter D is reported but it is never discussed and distinguished from NPQ. In material and methods it is stated: “Nonphotochemical quenching and its components are represented in the following equations: non-photochemical quenching (NPQ) = Fm /Fm’ – 1 and D = 1- (Fv’/Fm’).  1- (Fv’/Fm’) is not a measure of an NPQ component. It indicates the fraction of light energy absorbed that is not used for photochemistry. It comprehends NPQ but also fluorescence emission and triplets quenching.  Authors should discuss in detail the difference between NPQ and D and what these differences represent

Response: The statement has been modified as suggested, and the revised one now reads as follow.

“The formula “NPQ = Fm /Fm’ – 1” stands for non-photochemical quenching, which is divided into photoprotection and photoinhibition. When the plant is below the saturation luminosity, Pn will rise and maintain the level. At this time, the NPQ rises to the effect of photoprotection. At high luminosity, Pn will decrease, and the NPQ rises to the effect of photoinhibition. In addition, when the total energy is treated as 1, the energy quenching can be divided into P, D, and E, where P = (Fm'- Fs) / Fm' that is the ratio of photochemical quenching to absorbed energy; D = 1 - Fv'/ Fm' = (Fm' – Fo') / Fm which is the ratio of thermal quenching to absorbed energy; E =1 - P – D that is the ratio of excess energy to absorbed energy.”   (Line 428-437)

Line 69: “When mangrove ecosystems are exposed to high illumination and high salinity conditions, the plants increase solar energy absorption and decrease stomatal conductance, resulting in reduced CO2 fixation to generate excess energy.” This sentence is misleading; it’s not correct that in high light and reduced CO2 more excess energy is generated. In high light and reduced CO2, the electron transport became saturated and thus the light energy in excess could generate photooxidative stress.

Response: The statement “When mangrove ecosystems are exposed to high illumination and high salinity conditions, the plants increase solar energy absorption and decrease stomatal conductance, resulting in reduced CO2 fixation to generate excess energy.” has been changed. The revised one now reads as follow.

“When mangrove ecosystems are exposed to high illumination and high salinity conditions, the environmental stresses may decrease stomatal conductance, resulting in reduced CO2 fixation to generate excess energy.”  (Line 71-73)

Minor

Line 42-49: salinity is provided with three different units: percentage, millimole and part per thousand. Please use a common unit.

Response: We correct the salinity unit, which is expressed in part per thousand (ppt).

“Sabine et al. [6] revealed that mangroves have an optimal salinity range of 8 – 18 part per thousand (ppt), but Krauss et al. [7] indicated an optimal seawater con-centration range of 1.75 – 26.25 ppt, and the optimal salinity varies with species. Moreover, micro-tidal wetlands also show strong seasonal soil salinity variations (0 – 60 ppt) that may increase in amplitude according to climate prediction models, affecting the morphology and physiology of mangrove seedlings, thereby influencing the growth and species composition of mangrove swamps [4]. Previous studies have shown that salinities of 0 - 10 ppt are close to optimal for the growth of most mangroves [7, 8].”  (L41-49)

Line 67-68: “the photosynthesis of mangroves can reach light saturation under the incident photon flux density of 40% sunlight or lower”. Please provide an approximate value, of the light intensity that saturate photosynthesis in mangrove.

Response: We have added photometric units to Line 68-70 as suggested.   .

“In the harsh tropical intertidal environment of mangroves, the photosynthesis of man-groves can reach light saturation under the incident photon flux density of 40% sunlight (800-1,000 μmol photon·m-2·s-1) or lower [22, 23, 24, 25].”

Reviewer 4 Report

1. Based on the part of the Manuscript (lines 90-97), a specific research goal should be formulated in a separate sentence.

2. How was PPED controlled (lines 130-131)?

3. What radiation sources were used and why?

4. The formulas in lines 156 and 164 are very inconvenient to perceive inside the text. It is advisable to put each formula on a separate line.

5. For figures 1-5, it is advisable to use different colors.

Author Response

Based on the part of the Manuscript (lines 90-97), a specific research goal should be formulated in a separate sentence.

Response: We added sentences with specific research goals to Line 95-100 as suggested.

“Therefore, we designed three salinity and immersion experiments for K. obovata and R. stylosa seedlings. We aimed to elucidate the photosynthetic performances of both species treated with salinity and immersion under various light irradiations. The obtained data could explain differences in the distribution of mangrove species in the microenvironment and provide environmental conditions for mangrove restoration in the feature.”  (Line 95-100)

How was PPED controlled (lines 130-131)?

What radiation sources were used and why?

Response: We revised the sentences for clearance and shown in Line 134-137.

“After salinity and immersion treatments, the PPFD was adjusted to 0, 5, 10, 15, 25, 50, 75, 100, 200, 400, 800, 1,200, 1,500, 1,800, and 2,000 μmol photon·m-2·s-1 in the leaf chamber to understand the radiant energy of the tested plants under different illumination intensities in the flow state. The artificial light source was provided by LED lights.”  (Line 134-137). “

The formulas in lines 156 and 164 are very inconvenient to perceive inside the text. It is advisable to put each formula on a separate line.

Response: Each formula of ChlF has been stated on a separate line (Line 150 – 159) as suggested.

The chlorophyll fluorescence parameters we used are as follows:

Fv / Fm = (Fm - Fo) / Fm.................. .(1)

Fv’ / Fm’ = (Fm’ - Fo’) / Fm’……......(2)

ΔF/Fm' = (Fm' - F) / Fm'....................(3)

ΦPSII = (Fm’- Fs) / Fm’.....................(4)

ETR = ΦPSII x 0.5 x 0.84 x PPFD......(5)

ETR/PG = ETR/ (Pn + R)…...…….....(6)

NPQ = (Fm / Fm’) -1…………….......(7)

D = 1- (Fv’/ Fm’)..................................(8)

(1)(2) The potential (Fv/Fm) and actual (Fv'/Fm') photosystem II efficiency (ΦPSII) were calculated as (Fm - Fo) / Fm and (Fm' - Fo') / Fm', respectively. Fo (Fo') and Fm (Fm') are the minimal and maximal fluorescence values of dark-adapted and during illumination leaves, respectively [34].

(3)(4)  Measured leaves were dark-adapted for 30 min before light-curve runs. The effective quantum yield of PSII (ΔF/Fm’) was obtained using the light-curve program, where actinic light intensity was increased over 2 min in fifteen steps, illuminating at least 15 min in each stage to make the photoreaction fully mani-fested. The following effective quantum yields were measured and recorded using the light-response curve program [35]. ΔF/Fm’ was calculated as (Fm’ - F) / Fm’, where F is the fluorescence yield of the light-adapted sample and Fm’ is the maximum light-adapted fluorescence yield when a saturating light pulse was superimposed.

(5) The apparent rate of the photosynthetic electron transport (ETR) of PSII was obtained as ETR = ΔF/Fm’ x PPFD x 0.5 x 0.84, where the factor 0.5 assumed equal excitation of both PSII and PSI.

(6)  The ETR correction factor of 0.84 was used because only a fraction of incident light is actually absorbed by the two PSs [28]. Gross photosynthesis (PG) is the value of Pn in summation with the dark respiration rate (Rd) obtained from the linear regression of photosynthesis to illumination measured from 0 - 100 µmol PPFD m-2 s-1. The ETR/PG ratio represents the state of energy distribution, and a higher ratio of ETR to photo-chemistry suggests there may be excess energy.

(7)(8)  Non-photochemical quenching and its components are represented in the following equations: non-photochemical quenching NPQ and D [36, 37]. (Line 151-182)

For figures 1-5, it is advisable to use different colors.

Response: We have corrected the colors of figures 1-5, and R. stylosa seedling now is represented in blue.